# Impact of Sex on Completion of Life-Saving Interventions for Severely Injured Patients: A Retrospective Cohort Study

**Doriane Deloye** [1], **Alexandra Nadeau** [1,2], **Amanda Barnes-Métras** [1], **Christian Malo** [1,3], **Marcel Émond** [1,2,3], **Lynne Moore** [4], **Pier-Alexandre Tardif** [1], **Axel Benhamed** [1,3], **Xavier Dubucs** [1], **Pierre-Gilles Blanchard** [1,3] and **Eric Mercier** [1,2,3],*

1 Centre de Recherche du CHU de Québec—Université Laval, Québec, QC G1J 1Z4, Canada; doriane.deloye@crchudequebec.ulaval.ca (D.D.); alexandra.nadeau@crchudequebec.ulaval.ca (A.N.); amanda.barnes-metras.1@ulaval.ca (A.B.-M.); christian.malo.med@ssss.gouv.qc.ca (C.M.); marcelemond1@me.com (M.É.); pier-alexandre.tardif@crchudequebec.ulaval.ca (P.-A.T.); axel.benhamed@chu-lyon.fr (A.B.); xavier.dubucs@gmail.com (X.D.); pierre-gilles.blanchard.1@ulaval.ca (P.-G.B.)
2 VITAM—Centre de Recherche en Santé Durable, Québec, QC G1J 1Z4, Canada
3 Département de Médecine Familiale et de Médecine D'urgence, Faculté de Médecine, Université Laval, Québec, QC G1V 0A6, Canada
4 Département de Médecine Préventive, Faculté de Médecine, Université Laval, Québec, QC G1V 0A6, Canada; lynne.moore@fmed.ulaval.ca
* Correspondence: eric.mercier@fmed.ulaval.ca

**Abstract:** Sex disparities in access and quality of care are well known for some time-sensitive conditions. However, the impact of sex on early trauma care remains unknown. In this study, we compared delays of completion of life-saving interventions (LSIs) between females and males among severely injured patients. This is a retrospective cohort study of all patients who consulted or were transported by ambulance in the emergency department (ED) of a level-one trauma centre following injury between September 2017 and December 2019 and for whom at least one LSI was performed. The list of LSIs was established by an expert consensus and included trauma team leader (TTL) activation, endotracheal intubation, chest decompression, blood transfusion, massive transfusion protocol, neurosurgery, spinal surgery, intestinal surgery, and spleen, liver and/or kidney angiography. A total of 905 patients were included. No significant statistical differences in the LSI delays were found when comparing females and males brought directly to the ED and transferred from another health care setting. Results of this study suggest that delays before completion of LSIs are similar for severely injured patients at our major trauma centre regardless of their sex.

**Keywords:** trauma; sex difference; life-saving interventions; mortality

## 1. Introduction

Trauma is a major cause of death and lifelong disability worldwide [1]. Among adults < 45 years old, almost half of deaths are related to a trauma [2], of which approximately 90% are secondary to traumatic brain injury (TBI) and/or hemorrhagic thoracoabdominal injuries [3]. Despite recent improvements in trauma care, early trauma-related mortality remains high. For instance, in-hospital mortality due to TBI and thoracoabdominal injuries varies between 7.0% and 26.0% throughout Canadian provinces [3]. The median time to death is 29 h in TBI, while 50% of all deaths due to hemorrhagic shock secondary to thoracoabdominal injuries happen within 2 h following the trauma [3]. Hence, rapid identification and management of the life-threatening injury is essential to reduce mortality and mitigate long-term morbidity for traumatized patients [4]. Delays between emergency department (ED) arrival and life-saving interventions (LSIs) are a well-established quality indicator metric in trauma centres [5]. Studying factors influencing adherence to evidence-based care is essential to improve quality of care in severely injured patients.

Previous studies have suggested that the patient's sex influences the access to healthcare and the quality of care provided, from the out-of-hospital environment to specialized inpatient treatments [6]. For instance, significant sex disparities in time-sensitive emergencies such as cardiac care have been reported, suggesting that females are less likely to undergo extensive cardiac workup [7–13] and to receive cardiac treatment such as aspirin, nitroglycerin and IV access [14] and cardiac catheterization after myocardial infarction [15], resulting in worse outcomes overall [9]. Furthermore, females are less likely than males to be admitted to an intensive care unit [6,8] and to receive intravenous tissue plasminogen activator for a stroke [16]. Compared to males, females also wait longer to receive analgesic and other opioids administration [17,18] and to have a non-contrast computed tomography (CT) [19]. In trauma patients, severely injured females are less likely to be transported directly to a major trauma centre [1,6] and to be transferred from a lower- to a higher-level trauma centre [5,20].

However, few studies have investigated sex disparities in diagnostic and therapeutic care in severely injured patients [5,21]. In this study, we compared delays to completion of LSIs between females and males in severely injured patients at our major trauma centre. We hypothesised that females would experience longer delays to interventions than males.

## 2. Materials and Methods

### 2.1. Research Ethics Approval

The Research Ethic Board of the CHU de Québec—Université Laval approved this study (project no. 2021-5269).

### 2.2. Study Design, Setting and Study Population

This is a retrospective cohort study conducted at l'Hôpital de l'Enfant-Jésus-CHU de Québec (Quebec City, QC, Canada), a level-one trauma centre with an annual census of approximately 65,000 patients (ED and in-patient). This adult level-one trauma centre serves the whole Eastern region of the province. A prehospital bypass protocol is in place to ensure that severely traumatized patients are transported to this site if the expected transport time is <60 min [22].

All traumatized patients who consulted or were transported by ambulance in the ED between September 2017 and December 2019 and for whom at least one LSI was performed during their visit were included. The list of LSIs was established by consensus by a multidisciplinary group of regional experts (prehospital medical director, ED physician, trauma team leader (TTL), trauma surgeon). Interventions were considered for inclusion if they were time-sensitive and required some level of trauma expertise. Included LSIs were TTL activation, endotracheal intubation, chest decompression, blood transfusion, massive transfusion protocol (MTP), neurosurgery, spinal surgery, intestinal surgery, and spleen, liver and/or kidney angiography. Potentially eligible patients were identified through our local trauma registry. Patients who suffered burns without any other trauma were excluded as well as patients for whom the TTL was activated, but no injuries requiring hospital admission were diagnosed.

For patients transferred from another hospital, delays were calculated with time zero defined as patient's arrival at the level I trauma centre regardless of which LSIs or protocols had been performed in the referring hospital.

### 2.3. Outcomes

The primary outcome was comparing delays until completion of LSIs between females and males among severely injured patients.

### 2.4. Data Origins and Extraction

Prehospital and in-hospital medical records were reviewed and data were extracted by two trained medical students (DD, ABM). Relevant data were collected using Redcap™ software. The review team followed a specific protocol that included prehospital and

inhospital variables. The prehospital medical records were used to extract injury character-istics, vital signs, interventions completed and the delays associated with each. In-hospital medical records were reviewed to collect patient demographic data, LSIs and non-vital interventions performed as well as the time of those interventions, ED disposition and in-hospital death. ReaScrib™ software (Logibec, QC, Canada) is a software that allowed medical staff to compile real-time data on each LSI performed in the resuscitation room. Data on completion of each LSI were obtained using ReaScrib™ software when they were available or nursing and/or medical notes when unavailable. Non-vital interventions were defined as wound repair, fracture reduction, analgesia and initial imaging such as echography, X-rays in resuscitation and CT.

*2.5. Statistical Analyses*

Delays were defined as time to intervention from patient's arrival in the ED. If the patient had more than one LSI performed, the analyses were conducted on the delays prior to each intervention individually. Therefore, a patient could be included more than once if they had multiple LSIs. Patients who were initially oriented to the resuscitation room and those who were non-initially oriented to the resuscitation room were analysed as two different subgroups. The primary outcome (delays until completion of LSIs) was presented as median differences and confidence intervals, and differences between delays in males versus females were assessed using a non-parametric Wilcoxon two-sample test with a two-sided $p \leq 0.05$ chosen a priori as statistically significant. Median differences between males and females were calculated using observed differences, and bootstrapped 95% confidence intervals were estimated using the 2.5 and 97.5 percentiles of 5000 replicates for each intervention. Categorical variables were presented using proportions, and continuous variables were presented using means (SD). All analyses were performed with ExcelTM (Microsoft Corporation, Redmond, WA, USA, 2016) and Statistical Analysis System software (v. 9.4, SAS Institute, Cary, NC, USA). As the profile and the needs of injured patients brought directly to our trauma centre from the field differs from those of patients transferred from the hospital, both groups were analysed separately [23].

**3. Results**

*3.1. Population Demographics and Clinical Outcomes*

A total of 905 patients were included with a mean age of 53.2 years old (SD 20.8), of whom 675 (74.6%) were males. Overall, 48% were brought directly to the level-one trauma centre, and 52% were transferred from another health care setting. Among the patients who were brought directly to the hospital, 29.5% were females, and 70.5% were males. Among the patients who were transferred from another hospital, 21.7% were females, and 78.3% were males.

The main trauma mechanisms for both females and males were falls (41.4%) followed by motor vehicle collisions (MVC) (32.5%). Overall, 59.9% were initially oriented in a resuscitation room, and a total of 40.1% were non-initially oriented in a resuscitation room. Following hospital discharge, 66.1% returned home, 18.4% were transferred to another hospital or a long-term facility and 15.5% died. Overall, females and males shared similar characteristics with the exception that females were more likely to be brought directly to the trauma centre than males ($p = 0.008$), but less likely to have sustained a penetrating trauma ($p = 0.001$ (direct) and $p = 0.008$ (transfer)) (Table 1).

**Table 1.** Population characteristics.

| | All | Males | Females | *p* Value |
|---|---|---|---|---|
| | N = 905 | N = 675 | N = 230 | |
| Origin of the patients, n (%) | | | | |
| Brought directly to the level-one centre | 434 (48.0) | 306 (45.3) | 128 (55.7) | 0.008 † |
| Transferred from another health care setting | 471 (52.0) | 369 (54.7) | 102 (44.4) | |
| Age, mean (SD) | 53.2 (20.8) | 52.5 (20.6) | 55.2 (21.2) | 0.08 ‡ |
| Trauma mechanisms, n (%) | | | | |
| Penetrating | 50 (5.5) | 47 (7.0) | 3 (1.3) | 0.001 † |
| Blunt other | 102 (11.3) | 87 (12.9) | 15 (6.5) | 0.008 † |
| Fall | 375 (41.4) | 271 (40.2) | 104 (45.2) | 0.008 † |
| Motor vehicle collisions (MVC) | 294 (32.5) | 208 (30.8) | 86 (37.4) | 0.7 † |
| Pedestrian | 39 (4.3) | 25 (3.7) | 14 (6.1) | 0.13 † |
| Bicycle | 45 (5.0) | 37 (5.5) | 8 (3.5) | 0.29 † |
| First destination upon arrival, n (%) | | | | |
| Resuscitation room | 542 (59.9) | 406 (60.2) | 136 (59.1) | |
| Monitored bed | 80 (8.8) | 61 (9.0) | 19 (8.3) | 0.80 † |
| Unmonitored bed | 270 (29.9) | 197 (29.2) | 73 (31.7) | |
| Other | 13 (1.4) | 11 (1.6) | 2 (0.9) | |
| Cardiac arrest, n (%) | 29 (3.2) | 21 (3.1) | 8 (3.5) | 0.83 † |
| First destination after ED care, n (%) | | | | |
| ICU | 305 (33.7) | 236 (35.0) | 69 (30.0) | |
| Intermediate-level care | 38 (4.2) | 29 (4.3) | 9 (3.9) | |
| Ward | 105 (11.6) | 74 (11.0) | 31 (13.5) | |
| Operation room | 402 (44.4) | 298 (44.2) | 104 (45.2) | 0.63 † |
| Angiointervention room | 20 (2.2) | 15 (2.2) | 5 (2.2) | |
| Death in the ED | 25 (2.8) | 17 (2.5) | 8 (3.5) | |
| Other | 10 (1.1) | 6 (0.9) | 4 (1.7) | |
| Final patient status, n (%) | | | | |
| Survivor | 765 (84.5) | 569 (84.3) | 196 (85.2) | 0.83 * |
| Death | 140 (15.5) | 106 (15.7) | 34 (14.8) | |
| Destination upon hospital departure, n (%) | | | | |
| Home | 598 (66.1) | 443 (77.9) | 155 (79.1) | |
| Long term care facility | 11 (1.2) | 8 (1.4) | 3 (1.5) | 0.88 † |
| Other Hospital | 156 (17.2) | 118 (20.7) | 38 (19.4) | |

† Fisher's exact test; ‡ pooled T-test; * Chi-Square test. ED: emergency department, ICU: intensive care unit, SD: standard deviation.

*3.2. Delays Prior to Life-Saving Interventions*

Among females and males brought directly to the ED (*n* = 434), the median delays for the following LSIs were (females vs. males): endotracheal intubation: 0 h 20 vs. 0 h 25 (*p* = 0.47, median difference: −0 h 04, 95% CI [−0 h 11; 0 h 08]); chest decompression: 1 h 26 vs. 1 h 20 (*p* = 0.63, median difference: 0 h 06, 95% CI [−0 h 31; 1 h 08]); administration of the first blood products: 0 h 27 vs. 0 h 31 (*p* = 0.83, median difference: −0 h 04, 95% CI [−0 h 17; 0 h 18]); spinal surgery: 20 h 27 vs. 19 h 03 (*p* = 0.42, median difference: 1 h 23, 95% CI [−4 h 59; 10 h 03]); neurosurgery and intestinal surgery: 2 h 55 vs. 3 h 10 (*p* = 0.44, median difference: −0 h 14, 95% CI [−1 h 45; 1 h 22]); and angiography: 1 h 58 vs. 2 h 26 (*p* = 0.50, median difference: −0 h 28, 95% CI [−1 h 29; 11 h 09]) (Table 2).

**Table 2.** Intervention delays for patients brought directly to the ED.

| Variables | Females | | Males | | Females vs. Males | |
|---|---|---|---|---|---|---|
| | N | Med | N | Med | Median Differences [95% CI] | *p* Value * |
| TTL activation | 35 | 0 h 21 | 86 | 0 h 22 | −0 h 01 [−0 h 17; 0 h 07] | 0.217 |
| Endotracheal intubation | 34 | 0 h 20 | 87 | 0 h 25 | −0 h 04 [−0 h 11; 0 h 08] | 0.467 |
| Chest decompression | 24 | 1 h 26 | 77 | 1 h 20 | 0 h 06 [−0 h 31; 1 h 08] | 0.630 |
| Blood transfusion | 33 | 0 h 27 | 61 | 0 h 31 | −0 h 04 [−0 h 17; 0 h 18] | 0.828 |
| Massive transfusion protocol | 10 | 0 h 26 | 22 | 0 h 18 | 0 h 08 [−0 h 12; 0 h 28] | 0.422 |
| Surgery | | | | | | |
| Spinal surgery | 17 | 20 h 27 | 48 | 19 h 03 | 1 h 23 [−4 h 59; 10 h 03] | 0.419 |
| Neurosurgery and intestinal surgery | 14 | 2 h 55 | 47 | 3 h 10 | −0 h 14 [−1 h 45; 1 h 22] | 0.438 |
| Angiointervention | | | | | | |
| All angio | 5 | 1 h 58 | 21 | 2 h 26 | −0 h 28 [−1 h 29; 11 h 09] | 0.501 |

* Wilcoxon two-sample test. *p*-value and median difference for the angiointervention are to be read with caution because N < 5.

Among females and males transferred from another health care setting (*n* = 471), the median delay for the following LSIs were (females vs. males): endotracheal intubation: 8 h 13 vs. 0 h 49 (*p* = 0.30); chest decompression: 2 h 03 vs. 2 h 40 (*p* = 0.45, median difference: −0 h 37, 95% CI [−2 h 17; 1 h 05]); administration of the first blood products: 0 h 51 vs. 1 h 14 (*p* = 0.93, median difference: −0 h 23, 95% CI [−1 h 17; 2 h 54]); spinal surgery: 18 h 45 vs. 17 h 45 (*p* = 0.99, median difference: 1 h 00, 95% CI [−6 h 38; 5 h 51]); neurosurgery and intestinal surgery: 5 h 21 vs. 3 h 28 (*p* = 0.32, median difference: 1 h 53, 95% CI [−1 h 16; 6 h 41]); angiography: 10 h 06 vs. 2 h 24 (*p* = 0.30, median difference: 7 h 42, 95% CI [−1 h 38; 16 h 39]) (Table 2).

Median delays were similar between females and males relative to all LSIs.

*3.3. Intervention Delays for Patients Initially Oriented in a Resuscitation Room*

Among all patients who were initially oriented in a resuscitation room, whether they were brought directly or transferred from another health care setting, time to completion of LSIs did not differ significantly by sex (Table 3). Similar results were found among all patients non-initially oriented in a resuscitation room whether they were brought directly to the ED or transferred from another health care setting (Tables 1 and 2).

**Table 3.** Intervention delays for patients transferred from another healthcare setting.

| Variables | Females | | Males | | Females vs. Males | |
|---|---|---|---|---|---|---|
| | N | Med | N | Med | Median Differences (95% CI) | *p* Value * |
| TTL activation | 15 | 0 h 17 | 60 | 0 h 31 | −0 h 14 [−0 h 30; 0 h 17] | 0.181 |
| Endotracheal intubation | 1 | 8 h 13 | 25 | 0 h 49 | | |
| Chest decompression | 8 | 2 h 03 | 35 | 2 h 40 | −0 h 37 [−2 h 17; 1 h 05] | 0.449 |
| Blood transfusion | 11 | 0 h 51 | 38 | 1 h 14 | −0 h 23 [−1 h 17; 2 h 54] | 0.934 |
| Massive transfusion protocol | 5 | 0 h 25 | 8 | 0 h 23 | 0 h 01 [−0 h 12; 0 h 09] | 0.774 |
| Surgery | | | | | | |
| Spinal surgery | 39 | 18 h 45 | 117 | 17 h 45 | 1 h 00 [−6 h 38; 5 h 51] | 0.995 |
| Neurosurgery and intestinal surgery | 18 | 5 h 21 | 62 | 3 h 28 | 1 h 53 [−1 h 16; 6 h 41] | 0.322 |
| Angio | | | | | | |
| All angio | 6 | 10 h 06 | 12 | 2 h 24 | 7 h 42 [−1 h 38; 16 h 39] | 0.296 |

* Wilcoxon two-sample test. *p*-value and median difference for endotracheal intubation were not calculated since the N is limited.

## 4. Discussion

The results show that median delays to intervention were similar between females and males for all LSIs studied in our ED. Hence, there seems to be no impact of sex on the delays prior to completion of critical trauma interventions such as endotracheal intubation, blood product administration and access to surgery.

Literature on sex disparities in timely completion of LSIs in the ED for trauma patients is scarce. A retrospective observational cohort study by Ingram et al. aimed to describe sex differences in efficiency measures related to timeliness of trauma care by studying time to angiography, laparotomy, and spinal fixation. For time to laparotomy, a significant *p* value was found despite no difference in median time by sex (*p* = 0.04) [5]. Our own findings are inconsistent with those of Ingram et al. since we found no sex disparities in timely completion of intestinal surgery either among patients brought directly to the ED (*p* = 0.35) or among patients transferred from another health care setting (*p* = 0.41). For time to angiography and spinal fixation, Ingram et al. found no statistically significant difference between females and males (angiography: *p* = 0.63; spinal fixation: *p* = 0.69), which is in line with our own results (angiography: *p* = 0.50 (direct), *p* = 0.30 (transfer); spinal surgery: *p* = 0.42 (direct), *p* = 0.99 (transfer)) [5].

Female patients are known to have higher rates of undertriage when compared to male patients [24]. Ingram et al. demonstrated that females have a significantly longer ED length of stay (LOS) than male patients [5]; therefore, it has been suggested that the increased delay in female trauma patient care may occur at injury identification [25] since results show that time to intervention is not significantly different. This is in line with prior studies that have found longer delay to imaging and longer ED LOS for female patients consulting for myocardial infarction and strokes [19,26–28]. McGann et al. found that female patients presenting with acute abdominal pain must wait 43 min longer until CT order (*p* = 0.0012), hence delaying their access to surgery [28]. Our lack of sex disparities before completion of abdominal surgery could potentially be explained by the similar waiting time to CT for both female and male trauma patients. Moreover, medical teams are aware that sex-based disparities influence the quality of care. However, further studies are required at our trauma centre and other hospitals.

We must consider the possibility that the lack of sex disparities in completion of LSIs is the result of prompt and appropriate ED triage for severely injured female and male patients at our trauma centre. Trauma triage is often based on multiple tools to help assess the severity of the injuries, which might minimize sex disparities. It has been found that medicine based on longstanding adherence to evidence-based guidelines can help mitigate

sex disparities in quality of care in emergency medicine [29]. Another study showed that in acute coronary syndrome care the use of stratification tools has the potential to reduce sex disparities [12]. Madsen et al. suggested that in stroke patients, ED triage protocols may be effective in minimizing sex disparities [27]. Furthermore, the lack of sex disparities in our ED may be partly explained by the characteristics of our population. Indeed, since we studied severely traumatized patients, their medical condition is critical and potentially life-threatening, making rapid intervention regardless of sex the priority to improve patient morbidity and mortality. Hence, in injured patients requiring LSIs, sex might be less impactful than in less critically injured patients.

Limitations of this study include its single-centre population and retrospective nature. Indeed, since our study was conducted at a high-level trauma centre, our results may not be representative of lower-trauma trauma centres. Our centre has specific procedures in place for trauma care, and approximately 65,000 patients visit the hospital yearly; therefore, the external validity of our results might be limited. However, our hospital covers the eastern territory of the Province of Quebec; therefore, patients coming to the ED are a good representation of the province's trauma cases. Sex-based inequality in access to healthcare and completion of LSIs may vary from one healthcare centre to another; therefore, the results of this study would most likely be different if it were conducted in another region. Furthermore, as it is a retrospective cohort study, missing data was predictable. Indeed, in the trauma and LSIs context, manuscript files are likely to be completed by the medical staff after the episode of care, which can lead to inconsistencies and unclear data. However, most of the data collected for the patients in the resuscitation room come from the ReaScrib ™ software, which collected data in real time during the episode of care, thus reducing the amount of missing data. Moreover, guidelines suggesting delays on timeliness of critical interventions are not available as literature on LSIs in trauma care is scarce; therefore, interpretation of delays in terms of timeliness should be done with caution. Finally, our limited study sample for certain interventions such as surgery and angiography may limit the validity of our findings, and results may be influenced by extreme values when the numbers are limited.

This study showed that, in our level-one trauma centre ED, delays to timely completion of multiple LSIs are similar between female and male trauma patients. These results respect the principle of equal access and quality of care for all patients on which the Canadian health care system is based [8]. According to the literature, errors in traumatic healthcare such as delayed operative or angiographic control of intrathoracic, abdominal, and pelvic hemorrhage along with longer delays for female trauma patients contribute to inpatient trauma deaths [20,24,30]. Since our findings did not reveal sex disparities, we could hypothesise that, at our ED, severely injured trauma patient outcome is less influenced by sex bias, though further investigations are required relative to sex-related difference in the whole continuum of care.

The findings of this study suggest that future studies are necessary to further examine potential sex disparities in completion of LSIs in the ED since there is currently no clear consensus on the subject. Moreover, no specific or clear reasons have been identified to explain the presence or absence of sex disparities. Hence, future research is needed to study possible explanations for such bias in patient care to help improve quality of care and patient outcome.

### 5. Conclusions

Although sex does not seem to influence time to life-saving intervention significantly between females and males for our severely traumatized population, patient's sex might impact the rest of the trauma care continuum differently for female and male patients. Further studying sex disparities in trauma care such as delays to other interventions like analgesia administration would help improve our healthcare system.

**Author Contributions:** Conceptualization, E.M., A.B.-M., A.N., C.M. and D.D.; methodology, E.M., M.É., L.M., P.-A.T. and A.N.; validation, A.B., X.D. and P.-G.B.; formal analysis, L.M., P.-A.T. and D.D. writing—original draft preparation, D.D., A.N. and E.M.; writing—review and editing, A.B.-M., C.M., M.É., L.M., A.B., X.D., P.-A.T. and P.-G.B.; funding acquisition, E.M. All authors have read and agreed to the published version of the manuscript.

**Funding:** This research was funded by Département de médecine familiale et médecine d'urgence, Faculté de médecine, Université Laval (internal grant) and Fondation du CHU de Québec (internal grant).

**Institutional Review Board Statement:** The study was conducted in accordance with the Declaration of Helsinki, and approved by the Institutional Ethics Committee of Centre de recherche du CHU de Québec—Université Laval (2021-5269).

**Informed Consent Statement:** Patient consent was waived due to the retrospective study design.

**Data Availability Statement:** Data could be available upon request.

**Conflicts of Interest:** The authors declare no conflict of interest.

## Appendix A

**Table 1.** Intervention delays for patients initially oriented in resuscitation room.

| Variables | Females | | | | Males | | | | Females vs Males | | p Value [1] | |
|---|---|---|---|---|---|---|---|---|---|---|---|---|
| | Brought Directly | | Transferred | | Brought Directly | | Transferred | | Median Differences [95% CI] | | | |
| | N | Med | N | Med | N | Med | N | Med | Brought Directly | Brought Directly | Brought Directly | Transferred |
| TTL activation | 34 | 0 h 19 | 13 | 0 h 11 | 84 | 0 h 21 | 54 | 0 h 28 | −0 h 01 [−0 h 16; 0 h 07] | −0 h 01 [−0 h 16; 0 h 07] | 0.191 | 0.201 |
| Endotracheal intubation | 34 | 0 h 20 | 0 | | 84 | 0 h 25 | 17 | 0 h 19 | −0 h 04 [−0 h 10; 0 h 08] | −0 h 04 [−0 h 10; 0 h 08] | 0.637 | |
| Chest decompression | 15 | 1 h 05 | 4 | 1 h 12 | 54 | 0 h 39 | 15 | 1 h 32 | 0 h 26 [−0 h 09; 0 h 47] | 0 h 26 [−0 h 09; 0 h 47] | 0.447 | 0.524 |
| Blood transfusion | 25 | 0 h 20 | 6 | 0 h 43 | 56 | 0 h 27 | 26 | 0 h 49 | −0 h 07 [−0 h 17; 0 h 04] | −0 h 07 [−0 h 17; 0 h 04] | 0.191 | 0.684 |
| Massive transfusion protocol | 9 | 0 h 22 | 5 | 0 h 25 | 22 | 0 h 18 | 8 | 0 h 23 | 0 h 04 [−0 h 16; 0 h 23] | 0 h 04 [−0 h 16; 0 h 23] | 0.520 | 0.774 |
| Surgery | | | | | | | | | | | | |
| Spinal surgery | 10 | 23 h 33 | 3 | 13 h 05 | 36 | 17 h 36 | 23 | 9 h 22 | 5 h 57 [−6 h 30; 41 h 57] | 3 h 43 [−9 h 00; 45 h 25] | 0.126 | 0.751 |
| Neurosurgery and intestinal surgery | 14 | 2 h 55 | 5 | 1 h 50 | 42 | 3 h 02 | 32 | 2 h 17 | −0 h 06 [−1 h 41; 1 h 33] | −0 h 27 [−1 h 22; 3 h 06] | 0.541 | 0.582 |
| Angio | | | | | | | | | | | | |
| All angio | 4 | 1 h 45 | 3 | 1 h 07 | 17 | 1 h 50 | 8 | 1 h 50 | −0 h 05 [−1 h 55; 4 h 58] | −0 h 43 [−1 h 50; 15 h 32] | 0.724 | 0.765 |

[1] Wilcoxon two-sample test.

**Table 2.** Intervention delays for patients non-initially oriented in resuscitation room.

| Variables | Females | | | | Males | | | | Females vs Males | | | |
|---|---|---|---|---|---|---|---|---|---|---|---|---|
| | Brought Directly | | Transferred | | Brought Directly | | Transferred | | Median Differences [95% CI] | | *p* Value [1] | |
| | N | Med | N | Med | N | Med | N | Med | Brought Directly | Transferred | Brought Directly | Transferred |
| TTL activation | 1 | 1 h 45 | 2 | 0 h 33 | 2 | 1 h 51 | 6 | 1 h 15 | −0 h 06 [−0 h 22; 0 h 09] | −0 h 41 [−1 h 55; 0 h 18] | 1 | 0.432 |
| Endotracheal intubation | 0 | | 1 | 8 h 13 | 3 | 2 h 15 | 8 | 7 h 06 | | 1 h 07 [−11 h 52; 5 h 22] | | 1 |
| Chest decompression | 9 | 3 h 20 | 4 | 2 h 52 | 23 | 4 h 43 | 20 | 4 h 28 | −1 h 23 [2 h 53; 2 h 10] | −1 h 36 [−4 h 50; 5 h 19] | 0.386 | 0.517 |
| Blood transfusion | 8 | 7 h 21 | 5 | 3 h 51 | 5 | 5 h 10 | 12 | 4 h 39 | 2 h 11 [−9 h 13; 12 h 09] | −0h48 [−7 h 45; 42 h 52] | 0.618 | 0.876 |
| Massive transfusion protocol | 1 | 0 h 55 | 0 | | 0 | | 0 | | | | | |
| Surgery | | | | | | | | | | | | |
| Spinal surgery | 7 | 18 h 51 | 36 | 19 h 04 | 12 | 24 h 12 | 94 | 18 h 52 | −5 h 21 [−15 h 21; 2 h 47] | 0 h 12 [−8 h 00; 5 h 06] | 0.135 | 0.607 |
| Neurosurgery and intestinal surgery | 0 | | 13 | 6 h 20 | 5 | 4 h 01 | 30 | 5 h 22 | | 0 h 57 [−4 h 03; 10 h 53] | | 0.555 |
| Angio | | | | | | | | | | | | |
| All angio | 1 | 13 h 35 | 3 | 16 h 13 | 4 | 3 h 02 | 4 | 3 h 20 | 10 h 33 [10 h 18; 10 h 49] | 12 h 53 [−0 h 26; 18 h 07] | 0.349 | 0.163 |

[1] Wilcoxon two-sample test.

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
