# Peer review of "Impact of Sex on Completion of Life-Saving Interventions for Severely Injured Patients: A Retrospective Cohort Study"

_traumacare, doi:10.3390/traumacare3040022_

Round 1
Reviewer 1 Report
This retrospective cohort study seeks to evaluate if females experience longer delays to life saving intervention than males in severe trauma. The key message is that sex does not influence time to life-saving intervention.
Strengths: Adds data to the gap in literature.
Weaknesses: Retrospective single center study, may only be applicable in similar mature trauma system settings. Does not differentiate in terms of trauma severity/ISS.
Overall: It is good to see that the results of this study show that there is no difference in time to life-saving intervention based on gender whether they presented directly to emergency, or were transferred from another hospital. As such, all patients are getting timely first interventions based on the severity of their trauma, rather than their gender. It would be interesting to see if there were any delays with subsequent interventions after the first one was attended to.
Comments:
1. The authors interchanged reanimation and resuscitation repeatedly throughout paper.
2. P7L199: This makes sense because the primary and secondary surveys do not differentiate on gender.
3. P7L208: This line sums up most people’s views.
4. Was there any difference in higher vs lower injury severity?
Author Response
Thank you for your review. We agree with your comments. The impact of sex on delays relative to the whole continuum of care would be interesting to study. It would require a different analytic approach, but we are considering it for another manuscript.
- The authors interchanged reanimation and resuscitation repeatedly throughout paper.
Response: Thank you for pointing this out. We have replaced reanimation for resuscitation everywhere in the paper.
- P7L199: This makes sense because the primary and secondary surveys do not differentiate on gender.
Response: Thank you for this comment. We agree.
- P7L208: This line sums up most people’s views.
Response: Thank you for this comment.
- Was there any difference in higher vs lower injury severity?
Response: Thank you for asking, it would have indeed been a great information to have. However, we haven’t calculated the injury severity score (ISS), or any other severity score, for our cohort due to resource constraints. Given the large number of patients included and the required expertise to generate ISS, we were unable to calculate the ISS for every patient.
Reviewer 2 Report
This is a well-written manuscript. As such I have only minor comments.
- Noting numbers of patients who died prior to receiving any intervention (and thus may be included in the overall numbers, but not the tables in the results) would be useful.
- From my reading, the numbers in 2 and 3 represent the median time to first intervention for all patients. The numbers do not align with what is presented in the text for the total N of patients in each group. For example, the text states that the total number brought to the ED is 434, but the sum of patients included in Table 2 adds to over 600. Each patient can only have one first event, so I am wondering exactly what is represented there.
- The study clearly states and acknowledges its purpose and design. However as a single-centre study, conducted at a major trauma centre, it is important to acknowledge that this may not be representative of other trauma centres. It should be well-emphasised that this could be the case. The authors have done this in the discussion; some more information about the centre could be provided to help the reader identify how applicable these results might be. Top trauma centres often have excellent training and procedures in place which might not be replicated across all trauma centres, and some acknowledgement of this may be useful.
- The results are not further explored, e.g. by injury subtype. I do wonder whether adjusting for this, or conducting analyses by trauma subtype might also be useful. However, this is just an observation; there probably is already enough in this as a stand-alone, descriptive, piece.
- The angio results are interesting- since other studies have found a difference between males and females in cardiovascular interventions, and possibly with a bigger sample size this may differ in the patients who were transferred. This would align with previous literature finding sex differences in treatments for acute coronary syndromes (e.g. bmjopen.bmj.com/content/9/7/e026507). However, these studies are in quite different settings.
- the median difference and 95% confidence intervals would be useful to include in the results section instead of p-values only, particularly given the small patient numbers for some comparisons.
Author Response
Thank you for your review and kind comments.
- Noting numbers of patients who died prior to receiving any intervention (and thus may be included in the overall numbers, but not the tables in the results) would be useful.
Response: Thank you for this suggestion. We agree that it would have been relevant and interesting. However, we did not collect the time of death during the data extraction process. Hence, we only know if the patient died or survived during the episode of care. Hence, we can’t calculate the number of patients who died prior to receiving any intervention. We would need to go back and extract this variable for every patient in our database to calculate the number of patients who died prior to receiving any intervention which would be impossible due to resource constraints and lack of funding.
- From my reading, the numbers in 2 and 3 represent the median time to first intervention for all patients. The numbers do not align with what is presented in the text for the total N of patients in each group. For example, the text states that the total number brought to the ED is 434, but the sum of patients included in Table 2 adds to over 600. Each patient can only have one first event, so I am wondering exactly what is represented there.
Response: Thank you for pointing this out. Apologies for the confusion. Tables 2 and 3 present the median time to every intervention, not just the first one. The N in the tables represent the number of interventions performed and not the number of patients. Hence, one patient could be in more than one intervention, which is why the total in the tables is much larger than the number of patients.
We chose this analytic approach for a few reasons. Notably, if we presented only the results for the median time to the first intervention only, we would have introduced a hierarchy and make a choice relative to the importance of one intervention versus another. We don’t think it would have been representative of the reality. In high-level trauma centres, critical interventions can often be performed concomitantly (or almost).
Unfortunately, an old sentence remained in this version of the manuscript and was overlooked during our review process. This sentence was removed for the article:
If the patient had more than one LSI performed, the analyses were conducted only regarding delay until the first intervention.
It was replaced by this sentence:
If the patient had more than one LSI performed, the analyses were conducted on the delays prior to each intervention individually. Therefore, a patient can be included more than once in tables 2 and 3 if he had multiple LSIs.
- The study clearly states and acknowledges its purpose and design. However as a single-centre study, conducted at a major trauma centre, it is important to acknowledge that this may not be representative of other trauma centres. It should be well-emphasised that this could be the case. The authors have done this in the discussion; some more information about the centre could be provided to help the reader identify how applicable these results might be. Top trauma centres often have excellent training and procedures in place which might not be replicated across all trauma centres, and some acknowledgement of this may be useful.
Response: Thank you for this suggestion. We agree. This was added to the text:
Indeed, since our study was conducted at a high-level trauma centre, our results may not be representative of lower-level trauma centres. Our center has specific procedures in place for trauma care and approximately 65 000 patients visit the hospital yearly; therefore, the external validity of our results might be limited.
- The results are not further explored, e.g. by injury subtype. I do wonder whether adjusting for this, or conducting analyses by trauma subtype might also be useful. However, this is just an observation; there probably is already enough in this as a stand-alone, descriptive, piece.
Response: Thank you for this suggestion. However, we were afraid that performing subgroup analysis relative to the injury subtype and trauma mechanism (or others) would have increased the number of analysis and could have artificially produced statistically significant values for comparisons with a small number of patients (ex: three women with penetrating injury, eight women with bicycle injury). Based on this reasoning, our statistician (PAT) believes that subgroup analyses would not give us information.
- The angio results are interesting- since other studies have found a difference between males and females in cardiovascular interventions, and possibly with a bigger sample size this may differ in the patients who were transferred. This would align with previous literature finding sex differences in treatments for acute coronary syndromes (e.g. bmjopen.bmj.com/content/9/7/e026507). However, these studies are in quite different settings.
Response: Thank you for this comment. We were also interested by this difference in the literature. We, however, decided not to explore this in the discussion since the study population and the study settings are quite different.
- the median difference and 95% confidence intervals would be useful to include in the results section instead of p-values only, particularly given the small patient numbers for some comparisons.
Response: Thank you for pointing this out, we have added the median difference and 95% confidence intervals for Tables 2 and 3. These sentences were also added in the paper to reflect this change:
Materials and methods, statistical analyses: The primary outcome (delays until completion of LSIs) was presented as median differences and 95% confidence intervals…
Materials and methods, statistical analyses: Median differences between males and females were calculated using observed differences and bootstrapped 95% confidence intervals were estimated using the 2.5 and 97.5 percentiles of 5000 replicates for each intervention.
Discussion: Results may be influenced by extreme values when the numbers are limited.
Thank you very much again for your review.
Round 2
Reviewer 1 Report
all comments addressed.
Reviewer 2 Report
Thank you- all comments have been satisfactorily addressed.